# Immune Escape and Drug Resistance Mutations in Patients with Hepatitis B Virus Infection: Clinical and Epidemiological Implications

**DOI:** 10.3390/life15040672

**Published:** 2025-04-20

**Authors:** Maria Antonia De Francesco, Franco Gargiulo, Francesca Dello Iaco, Laert Zeneli, Serena Zaltron, Giorgio Tiecco, Simone Pellizzeri, Emanuele Focà, Arnaldo Caruso, Eugenia Quiros-Roldan

**Affiliations:** 1Institute of Microbiology, Department of Molecular and Translational Medicine, University of Brescia-ASST Spedali Civili, 25123 Brescia, Italy; gargiulof@gmail.com (F.G.); f.delloiaco@studenti.unibs.it (F.D.I.); arnaldo.caruso@unibs.it (A.C.); 2Unit of Infectious and Tropical Diseases, Department of Clinical and Experimental Sciences, University of Brescia and ASST Spedali Civili di Brescia, 25123 Brescia, Italy; l.zeneli@unibs.it (L.Z.); g.tiecco@unibs.it (G.T.); emanuele.foca@unibs.it (E.F.); eugeniaquiros@yahoo.it (E.Q.-R.); 3Unit of Infectious and Tropical Diseases, ASST Spedali Civili di Brescia, 25123 Brescia, Italy; zaltron.serena@tiscali.it; 4Department of Clinical and Experimental Sciences, University of Brescia, 25123 Brescia, Italy; simone.pelliz@gmail.com

**Keywords:** HBV, escape mutations, drug resistance mutations, genotypes, vaccine, sequence

## Abstract

Hepatitis B virus (HBV) genetic variability, shaped by high mutation rates and selective pressures, complicates its management and increases the emergence of drug-resistant and immune-escape variants. This study aims to analyze immune escape mutations (IEMs) and drug resistance mutations (DRMs) in patients with HBV infection exposed to antiviral therapies and exhibiting detectable plasma HBV viremia. This monocentric retrospective real-life study was carried out at the ASST Spedali Civili di Brescia, Italy, from 2015 to 2023. A total of 102 consecutive patients with detectable serum HBV-DNA exposed to at least one NA and for whom a drug resistance assay was available were included in our study. HBV sequences were amplified, sequenced, and analyzed for mutations using Geno2pheno and Stanford University tools. Phylogenetic analysis and statistical regression were performed to confirm genotypes and identify mutation patterns and associated risk factors. Our study shows a 38.2% prevalence of DRMs, with M204I/V (95%) and L180M (64%) being the most common, and a 43% prevalence of IEMs, primarily in the major hydrophilic region. Genotype D3 exhibited a higher mutation burden than other genotypes. Significant associations were found between HBsAb presence and increased IEM burden, while HBeAg was protective against DRMs. Atypical serological profiles were observed in 18.6% of patients, including cases of HBV reactivation under immunosuppressive therapy. This study highlights the high prevalence of IEMs and DRMs in a real-world setting, particularly among HBV genotype D3 carriers. These findings underscore the importance of mutation surveillance to guide therapeutic strategies, vaccine design, and public health policies to address the challenges posed by HBV genetic variability.

## 1. Introduction

Hepatitis B virus (HBV) infection continues to represent a critical global health challenge, contributing to significant morbidity and mortality. The World Health Organization (WHO) has set an ambitious goal to eliminate HBV by 2030 [1]. Currently, approximately 296 million individuals globally are living with chronic HBV infection [2], with an estimated prevalence of 0.9% in Europe [3].

The nationwide vaccination program introduced in Italy in 1991 for all newborns and 12-year-old children, associated with better socio-economic conditions and the adoption of public health interventions, led to a significant decline in HBV infection rates [4]. However, even if the overall prevalence of HBV in Italy is approximately 0.6% [4], the disease’s impact is still considerable, in particular for the increasing number of not vaccinated immigrants from high-risk regions [4].

Then, a great reduction in the impact of major risk factors for HBV infection acquisition occurred. Vertical transmission from HBsAg positive mother is a rare event, due to the obligatory determination of serum HBsAg in pregnant women [5]. So far, sexual transmission is principally responsible for HBV transmission in Italy, due to condom nonuse in unsafe sexual activity [5].

The high error rate introduced by reverse transcriptase during HBV replication results in frequent nucleotide substitutions, leading to genotype and subgenotype diversity [6]. The HBV mutation rate is estimated to range between 10^−4^ and 10^−4^ substitutions per site per year. Furthermore, the recombination events occurring during the virus replication give rise to a multitude of HBV variants that are closely linked genetically and constitute quasispecies. To date, HBV is classified, on the basis of genome nucleotide variation, into 10 genotypes and more than 35 subgenotypes [6]. These genetic variations, shaped by immunological and host-mediated pressures as well as antiviral therapies, play a crucial role in disease progression, transmission dynamics, and therapeutic response [7,8,9].

The mutation landscape of HBV has become a dynamic and complex area of investigation within virology, clinical medicine, and public health due to its profound impact on HBV pathogenesis, diagnosis, treatment, and prevention. A notable feature of HBV is the overlapping nature of its genome. Moreover, elective pressures from antiviral therapies, particularly those targeting the polymerase, together with the host immune response, which is principally directed to the surface antigen (HBsAg), contribute to the emergence of mutations that offer advantages in both therapeutic and immune evasion contexts.

The rise of drug-resistant variants is a pressing concern in HBV, especially in regions where entecavir (ETV), tenofovir disoproxil fumarate (TDF), and tenofovir alafenamide (TAF) are either unavailable or not fully reimbursed for treatment-naïve or treatment-experienced patients [10]. Mutations in the HBV polymerase gene, especially within regions encoding the catalytic domain, can lead to resistance against nucleos(t)ide analogs (NAs), which are the cornerstone of antiviral therapy. These mutations have the potential to undermine treatment efficacy, necessitating the implementation of alternative therapeutic strategies, even in the context of the availability of high-resistance-barrier agents with proven long-term effectiveness. Equally significant is the overlap between the polymerase and surface antigen genes, where mutations in the polymerase gene can result in alterations to the amino acid sequence of the HBsAg, potentially giving rise to immune escape variants. These immune escape variants can evade host immune responses, contributing to persistent infection, complicating serological diagnosis, and challenging vaccine development. Such variants may yield false-negative results in diagnostic tests and reduce vaccine effectiveness, representing a concern in public health. Moreover, HBV mutations significantly impact transmission dynamics and epidemiology. Variants with enhanced transmission fitness may spread more efficiently within populations, thus sustaining and propagating HBV infection on a global scale.

Few data are present in the literature about the prevalence of both HBV drug-resistant and immune escape mutations in Italy.

Therefore, we conducted a monocentric retrospective real-life study to analyze immune escape mutations (IEMs) and HBV drug resistance mutations (DRMs) in patients with HBV infection exposed to antiviral therapies and exhibiting detectable plasma HBV viremia.

## 2. Materials and Methods

### 2.1. Study Design and Participants

This study is a monocentric retrospective real-life analysis carried out at the ASST Spedali Civili di Brescia, Italy, from 2015 to 2023. The study included all consecutive patients with detectable serum HBV-DNA who have been exposed to at least one NA and for whom an HBV genotypic resistance test was available. No exclusion criteria were applied. HBV viral loads were determined by using the commercial assay COBAS HBV test (4800 System, Roche Diagnostics Italia, Monza, Italy) according to manufacturer’s instructions. Analysis was performed by the COBAS^®^ 4800 System Application Software (Version 2.1) and results were expressed in IU/mL. The detection limit was 10 IU/mL.

### 2.2. Definitions

The definitions used in this study are based on the sequential phases of chronic HBV infection as described in the 2017 European Association for the Study of the Liver (EASL) Clinical Practice Guidelines [10], as follows:-“Chronic hepatitis B needing antiviral therapy” (including phases #1, #2, and #4) refers to a condition in which patients are positive for HBsAg, have elevated liver enzyme levels or high HBV-DNA loads, and meet the criteria for antiviral therapy and HBV DNA and liver c enzyme levels increased during antiviral therapy.-“Chronic HBV infection non needing antiviral therapy” (phase #3) refers to a condition in which patients are positive for HBsAg, have detectable HBV viremia, and normal liver enzyme levels-“Occult hepatitis B virus” (HBV) infection (OBI) (phase #5) refers to a condition in which patients are negative for both serum hepatitis B surface antigen (HBsAg) and HBV-DNA, but positive for hepatitis B core antibody (anti-HBc), with or without detectable hepatitis B surface antibody (anti-HBs)

### 2.3. Data Collection

The clinical charts for all patients included in the study were reviewed, and relevant data were extracted, entered into an electronic database, and subsequently pseudo-anonymized. Collected information encompassed demographic characteristics (age, gender, country of origin), HBV serological markers (HBsAg, HBeAg, anti-HBe, and anti-HBs), liver markers (alanine aminotransferase, ALT, and aspartate aminotransferase, AST), presence of coinfections (HIV, HCV, and HDV), fibrosis-4 (FIB4) index, as well as the presence of cirrhosis and hepatocellular carcinoma.

### 2.4. Amplification of S/POL Region

HBV DNA was extracted by a QIAamp Blood Kit (Qiagen Italia, Milan, Italy) according to the manufacturer’s instructions. HBV genotyping and molecular characterization were performed by direct sequencing of the overlapped partial surface/polymerase (S/POL) region amplified by a nested PCR assay using primers described by Vincenti et al. [11]. For first-round PCR, the primers were POLHB1F (5-CCTGCTGGTGGCTCCAGTT-3) and POLHB2R (5-CRTCAGCAAACACTTGRC) nt 56–74 and nt 1175–1192 of HBV genome, respectively, while for the second round were POLHB3F (5-CTCGTGGTGGACTTCTCTC-3) and POLHB4R (5-GCAAANCCCMAAAGRCCCAC-3) nt 253–271 and nt 1000–1019 of HBV genome, respectively. PCR conditions were the following: denaturation 15 min at 94 °C, then 40 cycles of 1 min at 94 °C, 1 min at 58 °C, and 2 min at 72 °C.

### 2.5. Sequencing and Analysis of the Sequences

PCR products were purified with the QIAGEN Qiaquick PCR Purification Kit (Qiagen) according to the manufacturer’s instructions and direct nucleotide sequencing reaction was performed using Big Dye Terminator kit version 3.1 (Applied Biosystems, Foster City, CA, USA) with the primers used for the second-round PCR. Sequencing reactions were performed on an automated SeqStudio Genetic Analyzer (Applied Biosystems). Consensus sequences of each HBV isolate were submitted to a web-based tool for the prediction of clinically relevant mutations associated with drug resistance (Geno2pheno [hbv] 2.0, Max-Planck-Institut fur Informatik, Germany, available at http://hbv.geno2pheno.org/index.php, accessed on 1 August 2009) and a version developed by the Stanford University (available at HBV-drug resistance interpretation, accessed on 1 August 2009). The principal mutations associated with drug resistance are reported in Table 1.

Immune escape mutations were identified in the major hydrophilic region (MHR, aa 99–169) by using the same tools.

### 2.6. Phylogenetic Analysis

HBV sequences were aligned using the CLUSTAL W program in MEGA version 10 software with default settings. A phylogenetic tree was constructed using the Maximum Likelihood method of the MEGA10 software with the Kimura-2 parameter model with 1000 bootstrap replicates to assess the reliability of the output tree [12,13]. Reference sequences of HBV genotypes A-J and relative subtypes were downloaded from the HBV database (https://hbvdb.lyon.inserm.fr/HBVdb/HBVdbIndex, accessed on 1 August 2009). The Newick version of the phylogenetic tree constructed in MEGA 10 was imported into the Interactive Tree of Life (iTOL) for annotation and presentation [14]. The HBV sequences generated in this study (101/102 because of the unavailability of one sequence) were deposited into the NCBI GenBank under accession number PQ349088 to PQ349188.

### 2.7. Statistical Analysis

Statistical analysis was performed in R. Data were expressed as a median (interquartile ranges [IQR]) for quantitative variables and as a percentage for qualitative variables. The χ^2^ test was used for qualitative data, while the Mann–Whitney test was used for continuous data. Only correlations with *p* < 0.05 were considered statistically significant. Logistic regression analyses were performed to evaluate factors correlated with the presence of escape mutations ≥ 3 and resistance mutations. The final models were defined using an algorithm that performs the process of selection of variables by sequential removal (backward) according to the Akaike Information Criterion (AIC). They start from an analysis that includes the following variables in the model as possible predictor candidates: gender, age, geographical origin, genotypes, coinfections, cirrhosis, hepatocarcinoma, AST, ALT, HBsAg, HBeAg, anti-HBe, anti-HBs, and FIB4.

## 3. Results

This real-life study included 102 patients, the majority of whom were male (66/102, 65%) and of Italian origin (73/102, 71%) (Table 2).

The median age of the cohort was 68.5 (IQR, 59–76) years, with the youngest patient being 29 years old and the oldest 96 years old. Fourteen patients were diagnosed with cirrhosis, and four with hepatocellular carcinoma, while 25% (22/87) of the patients had a FIB-4 score greater than >2.5 (Table 2). Elevated alanine aminotransferase (ALT) levels greater than 40 IU/mL were observed in 48/102 patients (47%). Almost half of the patients tested positive for HBeAg (50/94, 53%) (Table 2). Notably, nearly half of them (24/50, 48%), while viremic, were non-Italian (17 of whom were under 50 years of age).

Our study population was categorized into three distinct groups according to the sequential phases of chronic HBV infection described by EASL. Group A (*n* = 30) comprised patients with occult hepatitis B virus (phase #5), who started antiviral therapy as prophylaxis due to concomitant immunosuppressive therapy, or were in follow-up after prophylaxis stopping [15] and became HBV DNA positive. Group B (*n* = 13) consisted of patients with chronic HBV infection not needing antiviral therapy (phase #3), who initiated antiviral therapy such as prophylaxis as a consequence of concurrent immunosuppressive conditions. These patients were receiving antiviral prophylaxis, not based on standard treatment indications, but as a precaution against concurrent immunosuppressive conditions. Patients from these two groups (A and B) had performed an HBV genotypic resistance assay because HBV DNA and liver enzyme levels increased during antiviral prophylaxis or during follow-up after prophylaxis stopping. Finally, Group C (*n* = 59) included patients with chronic hepatitis B who met the criteria for antiviral therapy (phases #1, #2, and #4) and the HBV genotypic resistance assay was performed because HBV DNA and liver enzyme levels increased during antiviral therapy.

The most prevalent HBV genotype in our cohort was genotype D, which was observed in 78% of the patients (Table 2).

The phylogenetic analysis of the HBV sequences obtained from the studied samples (Figure 1) revealed the following distribution: 7 samples (7%) clustered within genotype A, comprising 2 samples from the A1 subtype and 5 from the A2 subtype; 1 sample (1%) from an Asian patient belonged to genotype B; 6 samples (6%) clustered within genotype C, including 1 sample from the C2 subtype and 4 samples from the C1 subtype, all of which were from Asian patients; 6 samples (6%) clustered within genotype E, with 5 patients originating from Africa and 1 from Italy. The majority of samples (80/102, 78%) belonged to genotype D, with various subclusters represented. Most of them were from Italian patients (65/80, 81%). The D3 subtype was the most prevalent, with 56 samples identified, followed by 11 samples from the D1 subtype and 7 from the D2 subtype. Two Italian patients were identified with subgenotype F1, a subtype that is almost exclusively found in South America.

Of the 102 patients, known HBV mutations associated with drug resistance in the polymerase gene were identified in 39 patients (38.2%). The median number of mutations per patient was 2 (IQR, 1–3). Mutations at position 204 were the most frequently observed (37/39, 95%), followed by mutations at position 180 (25/39, 64%). Among all HBV strains harboring resistance mutations, genotype D3 exhibited a significantly higher prevalence of resistance mutations (74.3% [29/39] vs. 25.6% [10/39], *p* = 0.036) and a higher median number of resistance mutations (2 [IQR, 1–3] vs. 1.5 [IQR, 2]) compared to other genotypes (Table 3).

Regarding mutations in HBsAg, the prevalence of amino acid substitutions in the major hydrophilic region (MHR, aa 99–169) was 43% (44/102), with a median number of mutations in the MHR per patient of 1 (IQR, 1–2). The prevalence of mutations in the immunodominant “a” region (ADR, aa 124–147) was 37.2% (38/102), with a median number of mutations per patient of 1.9 (IQR, 1–2.5). Notably, one-quarter of patients harbored HBV strains with more than three immune escape mutations (Table 4).

The subtype D3 exhibited a higher prevalence of immune escape mutations compared to other genotypes (26/56, 46.4% vs. 18/46, 39.1%), although this difference was not statistically significant (*p* = 0.057). No IEMs or DRMs were detected in 34 patients, distributed across genotypes D (n = 21), E (n = 5), A (n = 3), C (n = 3), B (n = 1), and F (n = 1).

Among patients belonging to group A, 4 out of 30 patients (13.3%) tested HBsAg-negative/anti-HBs-positive, but with detectable plasma HBV-DNA, (Table 5). Two of them harbored HBV with IEMs but no DRMs, one harbored HBV with DRMs but no IEMs, and one patient (#4 in Table 5) harbored HBV with neither known IEMs nor DRMs. However, this patient’s HBV polymerase region contained the mutations F122L, H126Y, Q130P, Y135S, Y257F, D263E, I266V, and V278I.

In this study, 14.7% (15/102) of patients exhibited atypical HBV serological profiles because they tested positive for both HBsAg and anti-HBs (Table 6). This group included six cases from group A (five of whom exhibited immune escape mutations, while one showed lamivudine resistance mutations), one case from group B (with the I26IT mutation identified as an IEM), and eight cases from group C (three of whom exhibited immune escape mutations).

Stepwise linear regression analysis was performed to identify variables associated with the presence of HBV strains harboring ≥ 3 IEMs or DRMs. The presence of anti-HBs (OR 12.03, 95% CI 2.82–59.91, *p* < 0.001) was significantly associated with an increased risk of harboring HBV with ≥3 IEMs. Additionally, the presence of HBsAg (OR 12.09, 95% CI 1.32–285.95, *p* = 0.050), anti-HBs(OR 3.92, 95% CI 1.19–14.37, *p* = 0.030), and Italian origin (OR 5.29, 95% CI 1.64–21.56, *p* = 0.010) were associated with a higher likelihood of harboring HBV with DRMs, while the presence of hepatitis B e antigen (HBeAg) was identified as a protective factor against DRMs (OR 0.25, 95% CI 0.09–0.63, *p* = 0.004).

## 4. Discussion

This retrospective, monocenter, real-life Italian study, which included HBV-viremic, drug-exposed patients revealed the highest prevalence of IEMs reported in the literature. Specifically, ≥3 IEMs were identified in 33.3% of patients, and ≥3 IEMs were present in 9.8%. The most frequent IEMs were D144A/E (22%), followed by G145R, Y134H/N/S/V, and P120A/L/Q/S/T (18% each). A significant majority (80/102, 78%) of patients harbored HBV genotype D and were of Italian origin (65/80, 81%). Among them, the most frequent subtype was D3, associated with a higher prevalence of resistance mutations (29/39, 74.3%).

Comparative studies from Europe, including a multicenter survey (years 1998–2012) and a recent Italian study, reported lower IEM prevalences of 22% [16] and 18% [17], despite similar genotype D predominance. Another monocenter analysis performed in Egypt (years 2018–2020), including 81% of HBV genotype D, found a prevalence of IEMs of 16.3% [18]. These studies did not explicitly address whether OBI or HBV reactivations were included. Then, because changes in the S protein can result in vaccine escape mutants and also in diagnostic escape mutations which result in false negative HBsAg testing, the observed differences in the frequency of IEMs may result from the type of analyzed escape mutations.

Genotype D, the most prevalent HBV genotype in Europe [19], is characterized by significant heterogeneity in its surface (S) and polymerase (P) genes [20]. This genetic variability likely contributes to a genotype D-dependent attenuation of antibody-neutralizing activity induced by HBV vaccines or by the virus itself, with critical implications for public health [21]. Within the major hydrophilic region (MHR) of the HBV surface antigen, mutations at positions 126, 133, and 145 are considered the classic IEMs [22], although additional point mutations are frequently observed in OBI patients [23,24]. Our multivariate analysis found that the presence of anti-HBs was associated with an increased accumulation of IEMs (≥3), suggesting that MHR mutations may lead to antigenic changes affecting both antigenicity and immunogenicity. Furthermore, in patients who are anti-HBs positive, these antibodies might be detected in excess respect to the HBsAg, indicating a stronger immune response from the individual, which might be then responsible for the immune escape phenomenon.

Approximately 40.7% of patients in our study experienced HBV reactivation while receiving concurrent immunosuppressive and antiviral therapy, including OBI reactivations (group A) (29.4%) and chronic HBV infection reactivations (group B) (12.7%). Patients with OBI may test HBsAg-negative in plasma despite positive blood or intrahepatic HBV DNA, often due to S region mutations affecting HBsAg antigenicity or secretion. An analysis of 30,842,794 Italian blood donations (years 2009–2018) identified 1378 HBsAg-negative but HBV DNA-positive donors, 96.9% of whom had OBI [25]. Another Italian study on blood donors reported an alarming 75% frequency of IEMs in OBI patients, underscoring the public health risks posed by OBIs harboring transmissible IEMs, especially to HBV-vaccinated individuals [26,27], even if this represents at the moment a rare event. In our study, 63.3% (19/30) of OBI patients carried IEMs.

The true prevalence of OBI in Europe remains uncertain. A recent meta-analysis estimated a prevalence of 34%, but these data primarily reflect populations with chronic liver disease rather than the general population [28]. Given the high frequency of IEMs in OBI patients and consequently of OBI-associated variants, these individuals may serve as a reservoir for HBV strains with the potential of infecting vaccinated individuals.

In our study, the prevalence of DRMs was 38.2% and mostly associated with the D genotype. The most common DRMs were M204I/V (95%) and L180M (64%). This is somewhat lower than findings from European studies, such as Colagrossi et al., which reported a DRM prevalence of 54% (M204I/V: 46%, A181T: 2%) [13], and Hermans et al., which reported 52% (M204I/V: 48.7%, A181T: 3.8%) [29]. These discrepancies highlight that not all HBV-viremic patients receiving therapy harbor known DRMs. Furthermore, differences in the frequencies of drug resistance mutations among different studies might be due to the prevalence of different genotypes.

Then, from this dataset, it is difficult to determine whether drug resistance mutations are effectively more prevalent in genotype D infection, or if this simply reflects enrichment of genotype D in our country.

Multivariate logistic regression analysis showed that the presence of HBsAg, anti-HBs, and Italian origin were independent factors associated with greater risk for acquiring drug resistance mutations (OR = 12.09, *p* = 0.05, OR = 3.92, *p* = 0.03 and OR = 5.29, *p* = 0.01, respectively), underlining that HBsAg mutations can overlap with mutations in the reverse transcriptase (RT) region of HBV, leading to the appearance of drug resistance variants and to a selection of immune escape mutants which might occur in spite of the production of anti-HBs.

Furthermore, it confirmed the link observed between genotype D, higher frequency of drug resistance mutations, and Italian origin.

On the contrary, HBe antigen positivity was associated with a reduced risk of acquiring drug resistance mutations (OR = 0.25, *p* = 0.004), indicating that the virus is in a more active replication phase where probably the immune system exerts a smaller selective pressure leading to a lower number of mutations or that these patients might have a shorter infection time, reducing the number of virus mutations.

We found coexistence of HBsAg and anti-HBs in 14.7% of our patients, a percentage higher than those found in the literature where these atypical double-positive serological profiles were reported with percentages ranging from 2.8 to 3.6% among different cohorts [30]; then, they were correlated to the development of hepatocellular carcinoma [30].

A noteworthy observation in our study was that all six patients with HBV genotype E, who were on therapy, were viremic despite the absence of known DRMs. Genotype E, predominantly found in West Africa, is implicated in nearly 20% of chronic HBV infections worldwide. While data on its clinical outcomes and treatment efficacy are limited, genotype E is suggested to have a high carcinogenic potential and a lower response to antiviral treatment, often requiring prolonged therapy compared to other genotypes [31,32,33].

Our findings must be considered in the context of certain limitations. The retrospective design of this study introduces potential selection bias, and the cross-sectional nature precludes longitudinal observations. Moreover, no information about the duration of antiviral therapy, compliance of patients in assuming it, or type and duration of immunosuppressive therapies is available. Despite these limitations, this study provides a detailed examination of HBV immune escape and drug-resistance mutations in a real-life clinical setting.

## 5. Conclusions

This study underlines a high prevalence of IEMs and DRMs in a real-world setting, particularly among HBV genotype D3 carriers. Patients harboring HBV strains with IEMs could represent a significant risk for transmission, as the potential of vaccination to confer no protection against these mutated viruses is reported in many papers [16,17,34].

Understanding the genetic diversity of HBV is critical for developing tailored diagnostic, therapeutic, and preventive strategies to address the global burden of HBV-related morbidity and mortality effectively.

However, more studies are necessary to understand the clinical and epidemiological impact of IEMs and DRMs in the real world and might be useful to inform monitoring protocols and treatment strategies.

## Figures and Tables

**Figure 1 life-15-00672-f001:**
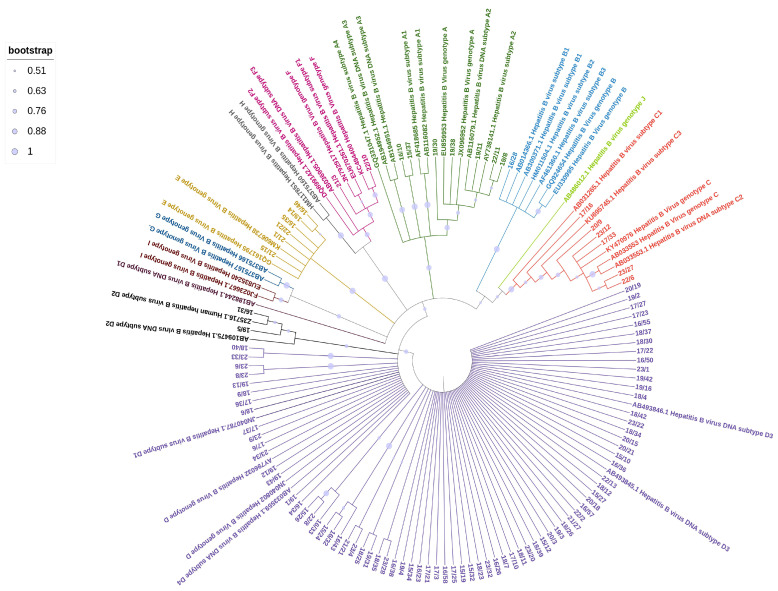
Maximum Likelihood tree of HBV sequences based on the alignment of 509 nucleotides of the surface/polymerase region along with 43 reference sequences from HBV genotypes and subgenotypes. Reference sequences are represented by the accession number and HBV genotype and subgenotypes.

**Table 1 life-15-00672-t001:** Principal mutations associated with drug resistance.

HBV (DRM)	Lamivudine	Entecavir	Adefovir	Tenofovir
Wild type	Susceptible	Susceptible	Susceptible	Susceptible
M204V/I	Resistant	Intermediate	Intermediate	Susceptible
L180M + M204V	Resistant	Intermediate	Intermediate	Susceptible
A181T/V	Resistant	Susceptible	Resistant	Susceptible
N236T	Susceptible	Susceptible	Resistant	Intermediate

**Table 2 life-15-00672-t002:** Clinical characteristics of the study population (*n* = 102).

Gender	
Female (%)	36 (35)
Male (%)	66 (65)
Age, median (IQR)	68.5 (59–76)
Country	
Italy, *n* (%)	73 (71)
Eastern Europe, *n* (%)	12 (12)
Africa, *n* (%)	7 (7)
Asia, *n* (%)	8 (8)
South America, *n* (%)	2 (2)
HBV DNA median (IQR) copies/mL	16,130,170 (726–488,162)
FIB4 index (*n* = 87)	
FIB4 index > 2.5, *n* (%)	22 (25)
HBeAg (*n* = 94)	
Positive, *n* (%)	50 (53)
Negative, *n* (%)	44 (47)
HBsAg (*n* = 102)	
Positive, *n* (%)	93 (91)
Negative, *n* (%)	9 (9)
HBsAg (*n* = 56) median (IQR) IU/mL	2153 (350–4889)
Anti-HBs (*n* = 93)	
Positive, *n* (%)	19 (20)
Negative, *n* (%)	74 (80)
ALT (*n* = 93)	
ALT median (IQR) IU/mL	44 (22.5–172.5)
AST (*n* = 93)	
AST median (IQR) IU/mL	37.5 (22–100)
HIV coinfection, *n* (%)	9 (9)
HCV coinfection, *n* (%)	0
HDV coinfection, *n* (%)	2 (2)
Patients with cirrhosis, *n* (%)	14 (14)
Patients with hepatocarcinoma, *n* (%)	4 (4)
NA treated patients (Group A) (*n* = 30)	
LAM treated	15 (14.7)
LAM + ENT treated	2 (2)
LAM + TDF treated	2 (2)
ENT treated	8 (7.8)
TDF treated	3 (2.9)
NA treated patients (Group B) (*n* = 13)	
LAM treated	7 (7)
ENT treated	4 (4)
TDF treated	2 (2)
NA treated patients (Group C) (n = 59)	
LAM treated	22 (21.5)
ENT treated	18 (17.6)
TDF treated	19 (18.6)
HBV genotypes	
Genotype A, *n* (%)	7 (7)
Genotype B, *n* (%)	1 (2)
Genotype C, *n* (%)	6 (6)
Genotype D, *n* (%)	80 (78)
Genotype E, *n* (%)	6 (6)
Genotype F, *n* (%)	2 (2)

Abbreviations: FIB4, fibrosis-4 index; LAM, lamivudine; ENT, entecavir; TDF, tenofovir disoproxil fumarate).

**Table 3 life-15-00672-t003:** Drug resistance mutations detected in the study.

Drug Resistance Mutations *	
Patients with drug resistance RT mutations (*n* = 39)	
First category	
Primary drug resistance mutations	N § (%)
M204I/V	37 (95)
S202G	2 (5)
A181T	1 (2.5)
T184A/S/I	3 (7.6)
N236T	0
I233V	0
M250V	3 (7.6)
Second category	
Compensatory mutations	
V173L/M	5 (12.8)
L180M	25 (64)
Third category	
Putative mutations	
S53N	0
T54N	0
L82M	0
Fourth category	
Pre-treatment mutations	
T38A	0
Y124H	0
D134E	0
New mutations in this study	
I169L/T	1 (2.5)
A181C	1 (2.5)
N236D	1 (2.5)
M250L	1 (2.5)
Median mutations (IQR)	2 (2)

Abbreviation: RT, reverse transcriptase; *, analyzed gene (RT); §, number of mutations.

**Table 4 life-15-00672-t004:** Immune escape mutations detected in the study population.

Immune Escape Mutations *	
Patients with immune escape mutations (*n* = 44)	
	N § (%)
Y100C	1 (2.2)
T118K	1 (2.2)
L109I/L	2 (4.4)
P120A/L/Q/S/T	8 (17.7)
T126I/N	3 (6.6)
P127A/L/T	6 (13.3)
A128V	3 (6.6)
Q129R	1 (2.2)
G130D/N/R	4 (8.8)
T131N	2 (4.4)
M133I/L/T	4 (8.8)
Y134H/N/S/V	8 (17.7)
P142L	2 (4.4)
D144A/E	10 (22.2)
G145R	8 (17.7)
G145A	1 (2.2)
Median mutations (IQR)	1 (1)
≥1 immune escape mutation, n (%)	34 (77)
≥3 immune escape mutations, n (%)	10 (23)

*, analyzed genes (pre-S/S); §, number of mutations.

**Table 5 life-15-00672-t005:** Patients with positive viremia, HbsAg-negative, and anti-HBs-positive.

Patient	Genotype	Escape Mutations	Mutations Associated with Drug Resistance	Previous Therapy	Current Therapy	HBV-DNA UI/mL	HbeAg	Anti-HBe	Group
1	D (D3)	D144DE	S	Lamivudine	Lamivudine	154	Neg	Neg	A
2	D (D3)	\	R (L180M, M204V), N236D unknown	Lamivudine	Lamivudine	265	Neg	Pos	A
3	D (D3)	P120Q, G145A	S	Lamivudine	NA	1140	Neg	Pos	A
4	D (D3)	\	S	Lamivudine	NA	2175	Neg	Neg	A

Abbreviations: Pos, positive; Neg, negative; NA, not available.

**Table 6 life-15-00672-t006:** Patients with positive viremia, HBsAg-positive, and anti-HBs-positive.

Patient	Genotype	Escape Mutations	Mutations Associated with Drug Resistance	Previous Therapy	Current Therapy	HBV-DNA UI/mL	HbeAg	HBeAb	Group
1	D (D3)	NA	R (204I), PR (204I)	NA	Lamivudine	1,685,000	Neg	Pos	A
2	D (D3)	128V, 129R, 130R, 134V	R (204I)	NA	Lamivudine	41,580,000	Neg	Pos	A
3	A (A1)	NA	R (180M,204V), PR (204V,180M)	NA	Lamivudine	1020	Neg	Pos	A
4	D (D3)	NA	R (204I)	NA	Lamivudine	97,772	Neg	Pos	A
5	D (D3)	142L, 144E, 145R	R (173L,180M,204V), PR (204V,180M)	Lamivudine	Tenofovir	512	Pos	Neg	C
6	D (D3)	NA	R (L180M, M204V, 169T,184A, 204V,180M)	Lamivudine, adefovir, tenofovir	Entecavir	695	Neg	Pos	C
7	D (D3)	P120L, M133MT, P142L, G145R	S	NA	Entecavir	726	NA	NA	C
8	D (D1)	NA	R (L180M, M204V, T184S)	NA	Entecavir	4287	Pos	Neg	C
9	D (D3)	T123P, S143L	S	Entecavir	NA	721	Pos	Neg	A
10	E	T126I, D144E, G145R	S	NA	Entecavir	325,000	Pos	Neg	A
11	D (D3)	Y134H	S	NA	Entecavir	23,800,000	Pos	Pos	A
12	D (D3)	142L, 144E, 145R	R (173L,180M,204V), PR (204V,180M)	Lamivudine	NA	18,980	Pos	Neg	A
13	D (D1)	144A, 145R	S	Lamivudine	NA	22,740,000	Pos	Neg	A
14	A (A2)	NA	S	NA	Lamivudine	559	Neg	Pos	A
15	C (C1)	I126IT	S	Entecavir	NA	70,700	Pos	Neg	B

Abbreviations: Pos, positive; Neg, negative; NA, not available.

## Data Availability

The data that support the findings of this study are available from the corresponding author upon reasonable request.

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
