# Peer review of "Immune Escape and Drug Resistance Mutations in Patients with Hepatitis B Virus Infection: Clinical and Epidemiological Implications"

_life, 2025, doi:10.3390/life15040672_

Round 1
Reviewer 1 Report
Comments and Suggestions for Authors
Interesting, useful cohort study.
Please define IEM and DRM clearly and explicitly exactly how your Methods will determine their presence - it seems to be mostly inferred at the moment.
To help clinicians treating such patients, please add duration of therapy for the cases with DRMs
Please also add the gene locus for Tables describing DRMs (RT) and IEMs (pre-S/S) for non-specialist readers.
What practical advice, based on their findings, can the authors offer to clinical teams managing such patients?
Author Response
Please define IEM and DRM clearly and explicitly exactly how your Methods will determine their presence - it seems to be mostly inferred at the moment.
We used a web-based tool for prediction of clinically relevant mutations associated to drug resistance (DRMs) (Geno2pheno [hbv] 2.0, Max-Planck-Institut fur Informatik, Germany, available at http://hbv.geno2pheno.org/index.php) and a version developed by the Stanford University (available at HBV-drug resistance interpretation). The same tools have been used to determine immune escape mutations (IEMs) in the major hydrophilic region (MHR, aa 99-169).
To help clinicians treating such patients, please add duration of therapy for the cases with DRMs
Unfortunately many patients came to our Hepatologic Unit to perform HBV resistance test when HBV DNA becomes positive and many informations about the clinical history of the patients are frequently lacking such as the duration of antiviral therapy. We have included this as another limitation of our study.
Please also add the gene locus for Tables describing DRMs (RT) and IEMs (pre-S/S) for non-specialist readers.
Thank you, we have added the genes as required
What practical advice, based on their findings, can the authors offer to clinical teams managing such patients?
We have added a brief sentence at the end of the Conclusions section.
Reviewer 2 Report
Comments and Suggestions for Authors
The manuscript, titled "Immune Escape and Drug Resistance Mutations in Patients with Hepatitis B Virus Infection: Clinical and Epidemiological Implications," addresses the extensively studied topic of HBV genetic variability among chronically infected patients in a specific geographic region. Although it provides no new scientific insights, it presents a comprehensive picture of the genetic variability of HBV isolates from patients with chronic HBV infection undergoing antiviral therapy at a single centre in Italy.
Points to be regarded:
The mutational rate of HBV is very high for a DNA virus; however, it is not responsible for the existence of genotypes and subgenotypes, as these result from long-term viral evolution and viral-host interactions.
Anti-HBs or anti-HBc should be used instead of HBsAb or HBcAb to increase clarity.
ALT, AST and others are usually referred to as liver enzymes, not cytolytic enzymes.
Having an OBI reactivation group and an HBV reactivation group is very confusing because they are often considered to be the same term. Instead, use the EASL classification of chronic infection phases for all patients.
The classification of patients into groups A and B is unclear (page 6, lines 197-199).
The letters in Figure 1 are too small to be visible without a high level of magnification. Try making it larger or present a smaller number of isolates.
The term “escape mutation” used throughout the text and in some tables is meaningless. It must be either immune-, diagnostic- or vaccine-escape.
The sentence on page 9, line 243, is missing a part.
The S gene variation T127P is not a mutation but a subgenotype variation.
The patients shown in Table 5 do not differ from other patients with OBI infection. Their positive anti-HBs antibodies still classify them as patients with occult infection.
Page 11, lines 276-280: When comparing the frequencies of immune-escape mutations across different studies, the observed differences may be due to variations in interpreting what constitutes immune-escape mutations.
Page 11, 288-291: Another interpretation of the higher frequency of immune-escape mutations in patients who are anti-HBs positive is that these antibodies are detected when they are present in excess relative to the HBsAg, indicating a stronger immune response from the individual, which is then responsible for the immune escape phenomenon.
Page 12, 299-300: Although transmission of mutant strains to vaccinated individuals has been described in the literature, it is actually a very rare event. Many studies describe vaccine-escape mutations without evidence of their transmission to vaccinated individuals.
Page 12, 310-315: As previously described in the literature, differences in the frequencies of resistance-associated mutations among different studies stem from genotype-specific differences.
The frequency of resistance-associated mutations is directly related to the duration of therapy, but the paper does not provide data on this.
Comments on the Quality of English LanguageThe language needs attention in some places.
Author Response
The mutational rate of HBV is very high for a DNA virus; however, it is not responsible for the existence of genotypes and subgenotypes, as these result from long-term viral evolution and viral-host interactions.
We agree with the referee’s observation and we have modified the sentence in the text.
Anti-HBs or anti-HBc should be used instead of HBsAb or HBcAb to increase clarity.
Thank you, we have modified accordingly
ALT, AST and others are usually referred to as liver enzymes, not cytolytic enzymes.
Thank you, we have modified accordingly
Having an OBI reactivation group and an HBV reactivation group is very confusing because they are often considered to be the same term. Instead, use the EASL classification of chronic infection phases for all patients.
We thank you for your advice, now we have used the EASL classification for chronic infection phases to define the groups of our study (see material and methods and results)
The classification of patients into groups A and B is unclear (page 6, lines 197-199).
We have better defined groups A and B based on EASL classification of chronic HBV infection phases
The letters in Figure 1 are too small to be visible without a high level of magnification. Try making it larger or present a smaller number of isolates.
We have increased the character of letters and numbers in Figure 1
The term “escape mutation” used throughout the text and in some tables is meaningless. It must be either immune-, diagnostic- or vaccine-escape.
We have changed the term “escape mutations” in “immune escape mutations” both in the text and in the Tables
The sentence on page 9, line 243, is missing a part.
Thank you, we have corrected the sentence
The S gene variation T127P is not a mutation but a subgenotype variation.
Thank you, we have deleted this sentence
The patients shown in Table 5 do not differ from other patients with OBI infection. Their positive anti-HBs antibodies still classify them as patients with occult infection.
Thank you again for your consideration. Now we have better explained the characteristics of patients showed in table 5
Page 11, lines 276-280: When comparing the frequencies of immune-escape mutations across different studies, the observed differences may be due to variations in interpreting what constitutes immune-escape mutations.
We agree with the referee’s observation, but all the mentioned studies analyzed as in our paper the same Major hydrophilic region (MHR) (aa.99-169)
Page 11, 288-291: Another interpretation of the higher frequency of immune-escape mutations in patients who are anti-HBs positive is that these antibodies are detected when they are present in excess relative to the HBsAg, indicating a stronger immune response from the individual, which is then responsible for the immune escape phenomenon.
We agree with the referee’s opinion and this sentence has been added to the text.
Page 12, 299-300: Although transmission of mutant strains to vaccinated individuals has been described in the literature, it is actually a very rare event. Many studies describe vaccine-escape mutations without evidence of their transmission to vaccinated individuals.
We agree and we have corrected accordingly.
Page 12, 310-315: As previously described in the literature, differences in the frequencies of resistance-associated mutations among different studies stem from genotype-specific differences.
We agree, but the considered studies showed also a higher prevalence of genotype D as in our study
The frequency of resistance-associated mutations is directly related to the duration of therapy, but the paper does not provide data on this.
We agree, but unfortunately many patients came to our Hepatologic Unit to perform HBV resistance test when HBV DNA becomes positive and many informations about the clinical history of the patients are frequently lacking such as the duration of antiviral therapy. We have included this as another limitation of our study.
Round 2
Reviewer 2 Report
Comments and Suggestions for Authors
Some of the issues from the previous review were not addressed correctly:
“The mutational rate of HBV is very high for a DNA virus; however, it is not responsible for the existence of genotypes and subgenotypes, as these result from long-term viral evolution and viral-host interactions.”
The authors have again confused the high genetic variability of HBV, which gives rise to quasispecies, with the evolution-dependent variability responsible for genotypes and subgenotypes. I suggest they further analyse genotypes' origins and add more references on the subject.
“ALT, AST and others are usually referred to as liver enzymes, not cytolytic enzymes.”
This was not corrected in the text since it is mentioned multiple times in the Methods. However, the EASL guidelines, which are now cited, do not mention "cytolytic enzymes."
“Having an OBI reactivation group and an HBV reactivation group is very confusing because they are often considered to be the same term. Instead, use the EASL classification of chronic infection phases for all patients.”
Table 2 still contains the terms "OBI reactivation" and "HBV reactivation". There is no phase called HBV reactivation in EASL guidelines. The only phenomenon called HBV reactivation is when patients who have OBI enter the immunosuppressive state and have reactivation of viral replication. So, the terms are one and the same. Patients from group B, as explained in the Results, were in phase 3 and could enter phase 4 by EASL guidelines (if they are HBeAg negative and have HBV DNA again). I suggest using the same group definitions in the tables as in the text (groups A, B, C).
“The term “escape mutation” used throughout the text and in some tables is meaningless. It must be either immune-, diagnostic- or vaccine-escape.”
The title in Table 4 still has the term “escape mutations”.
What does N stand for in Tables 4 and 5? Is it the number of mutations, patients, or isolates? It should be clearer.
“The patients shown in Table 5 do not differ from those with OBI infection. Their positive anti-HBs antibodies still classify them as patients with occult infection”. Why are they shown in a separate table?
Table 6 is not readable at all in the way that is currently shown.
“When comparing the frequencies of immune-escape mutations across different studies, the observed differences may be due to variations in interpreting what constitutes immune-escape mutations.”
The problem here is not in the observed region, which is MHR in most cases, but in interpreting what is an immune-escape mutation, and this should be explained in the Discussion.
“As previously described in the literature, differences in the frequencies of resistance-associated mutations among different studies stem from genotype-specific differences.”
I suggest adding more references to the link between drug-resistant mutations and the viral genotype (currently, there are none in the present manuscript). Given that genotype D is predominant, the current results might be biased. The previous literature data on this link favoured genotype A as most inclined towards drug-resistant mutations.
“Although transmission of mutant strains to vaccinated individuals has been described in the literature, it is actually a very rare event. Many studies describe vaccine-escape mutations without evidence of their transmission to vaccinated individuals.”
The Conclusions still strongly emphasise the possibility of transmission of vaccine-escape mutants, which, based on the literature, is proven to be a rare event. I suggest toning down this conclusion.
Author Response
Some of the issues from the previous review were not addressed correctly:
“The mutational rate of HBV is very high for a DNA virus; however, it is not responsible for the existence of genotypes and subgenotypes, as these result from long-term viral evolution and viral-host interactions.”
The authors have again confused the high genetic variability of HBV, which gives rise to quasispecies, with the evolution-dependent variability responsible for genotypes and subgenotypes. I suggest they further analyse genotypes' origins and add more references on the subject.
We have further modified our sentence on the basis of what reported in the literature as indicated by the following examples:
“HBV is replicated by an error-prone polymerase through an RNA intermediate, leading its classification into 10 genotypes, several subgenotypes and diverse intra host viral variants called quasispecies” (Scientific Reports 2024, doi.org/10.1038/s41598-024-60900-2, already inserted as reference)
“Because of the spontaneous error rate of viral reverse transcriptase, HBV genome evolves with an estimated rate of nucleotide substitution at 1.4–3.2 × 10−5/sites/yr (Lau and Wright 1993). This unique replication strategy leads to the occurrence of various genotypes, subtypes, mutants, recombinants, and even quasispecies in the context of the long-term evolution of HBV (Hunt et al. 2000; Kao 2002, 2003; Kao and Chen 2006)”. (Cold Spring Harb Perspect Med 2015 doi: 10.1101/cshperspect.a021436)
“ALT, AST and others are usually referred to as liver enzymes, not cytolytic enzymes.”
This was not corrected in the text since it is mentioned multiple times in the Methods. However, the EASL guidelines, which are now cited, do not mention "cytolytic enzymes."
We apologize for the oversight and for not having changed the terms referring to transaminases in the indicated section. We have now modified them.
“Having an OBI reactivation group and an HBV reactivation group is very confusing because they are often considered to be the same term. Instead, use the EASL classification of chronic infection phases for all patients.”
Table 2 still contains the terms "OBI reactivation" and "HBV reactivation". There is no phase called HBV reactivation in EASL guidelines. The only phenomenon called HBV reactivation is when patients who have OBI enter the immunosuppressive state and have reactivation of viral replication. So, the terms are one and the same. Patients from group B, as explained in the Results, were in phase 3 and could enter phase 4 by EASL guidelines (if they are HBeAg negative and have HBV DNA again). I suggest using the same group definitions in the tables as in the text (groups A, B, C).
Thank you, we have used also in the Table 2 the same group definitions as in the manuscript (groups A, B and C)
“The term “escape mutation” used throughout the text and in some tables is meaningless. It must be either immune-, diagnostic- or vaccine-escape.”
The title in Table 4 still has the term “escape mutations”.
We apologize for the oversight and we have corrected the title
What does N stand for in Tables 4 and 5? Is it the number of mutations, patients, or isolates? It should be clearer.
N in tables 3 and 4 refers to the number of mutations and it has been added.
“The patients shown in Table 5 do not differ from those with OBI infection. Their positive anti-HBs antibodies still classify them as patients with occult infection”.
Why are they shown in a separate table?
We just analyzed in detail the clinical characteristics of patients with OBI infection but with detectable HBV DNA. We have specified better this part in the manuscript
Table 6 is not readable at all in the way that is currently shown.
Sorry, probably there was a problem with the formatting of the table and now it has been reinserted
“When comparing the frequencies of immune-escape mutations across different studies, the observed differences may be due to variations in interpreting what constitutes immune-escape mutations.”
The problem here is not in the observed region, which is MHR in most cases, but in interpreting what is an immune-escape mutation, and this should be explained in the Discussion.
We have added a brief paragraph about the observed differences in IEMs in the Discussion section
“As previously described in the literature, differences in the frequencies of resistance-associated mutations among different studies stem from genotype-specific differences.”
I suggest adding more references to the link between drug-resistant mutations and the viral genotype (currently, there are none in the present manuscript). Given that genotype D is predominant, the current results might be biased. The previous literature data on this link favoured genotype A as most inclined towards drug-resistant mutations.
Thank you, we have inserted a better clarification about this point
“Although transmission of mutant strains to vaccinated individuals has been described in the literature, it is actually a very rare event. Many studies describe vaccine-escape mutations without evidence of their transmission to vaccinated individuals.”
The Conclusions still strongly emphasise the possibility of transmission of vaccine-escape mutants, which, based on the literature, is proven to be a rare event. I suggest toning down this conclusion.
We have attenuated our conclusions as suggested